**Data Availability Statement:** Data cannot be shared publicly because of data integrity and security. Data are available from the Universitas

# Integrating sustainable Islamic social finance: An Analytical Network Process using the Benefit Opportunity Cost Risk (ANP BOCR) framework: The case of Indonesia

Tika Widiastuti[1], Anidah Robani [2]*, Puji Sucia Sukmaningrum[1], Imron Mawardi[1], Sri Ningsih[3], Sri Herianingrum[1], Muhammad Ubaidillah Al-Mustofa[1]

1 Department of Sharia Economic, Faculty of Economic and Business, Universitas Airlangga, Surabaya, Indonesia, 2 Institute of Technology Management and Entrepreneurship, Universiti Teknikal Malaysia Melaka (UTeM), Melaka, Malaysia, 3 Department of Accounting, Faculty of Economic and Business, Universitas Airlangga, Surabaya, Indonesia

☯ These authors contributed equally to this work.
* anidah@utem.edu.my

## Abstract

The utilization of Islamic social finance instruments is far behind what is expected. To realize its full potential, Islamic social finance instruments must be integrated. This study examined solutions and priority strategies for integrating sustainable Islamic social finance that could be implemented in the short and long term using the Benefit, Opportunity, Cost, and Risk (BOCR) framework, which includes six aspects: Governance, sustainable financing, institutional aspect, human resources, regulations, and supporting technology. This qualitative research employed the Analytic Network Process (ANP) method using the benefit, opportunity, cost, and risk analysis. The data were obtained mainly through focus group discussions and in-depth interviews with respondents among academics, practitioners, associations, regulators, and community leaders. The respondents were selected for their expertise and experience in the selected topic. The data were processed using the Microsoft Excel and Super Decision software. There are several findings worth considering from the analysis. First, the highest priority in integrating Islamic social finance aspects are human resources (0.97), regulation (0.86), and technology (0.76). Second, based on the short- and long-term analysis, financial integration through sustainable financing (0.01 and 1.44, respectively) and improving human resource quality through certification and training (0.01 and 1.56, respectively) is a priority solution and strategy to integrate sustainable Islamic social finance. Meanwhile, according to expert judgments, integrating national data (0.24) and optimizing technology use (0.18) are priority solutions and strategies. The findings emphasize the critical role of improving human resource quality to utilize technology, with experts identifying a national data integration as the most critical solution. As a result, relevant stakeholders are concerned about technology management training for Islamic philanthropic managers, with the goal of maximizing the potential of technology's growing role and adoption.

Airlangga Institutional Data Access / Ethics Committee (contact via email ekis@feb.unair.ac.id) for researchers who meet the criteria for access to confidential data.

**Funding:** This research grant is a Covid-19 Research Mandate with the contract number 1025/UN3.15/PT/2021. The authors would like to thank the Research and Community Service Institution (LPPM) of Universitas Airlangga for their financial assistance. The authors would also like to thank all academicians, regulators, associations, practitioners, and community leaders for their support in writing this research. The funders had no role in study design, data collection and analysis, decision to publish, or preparation of the manuscript.

**Competing interests:** he authors have declared that no competing interests exist.

# Introduction

Islamic Social finance (ISF), which includes zakat, *infaq*, *sadaqah*, waqf, and Islamic microfinance, is critical to promoting a long-term economic development [1–3]. Islamic social finance has a vital role in solving the problem of poverty [3–6], income inequality [7], global education [8, 9], health access [10], and various other socio-economic problems. The role of Islamic social finance instruments is not only in the majority Muslim countries, but also in minority Muslim countries, such as the role of zakat in providing supportive programs for the poor in Singapore [11], the role of Islamic social finance to support economic resilience in the UK [12], and the role of zakat in creating prosperity in Germany, France, and Croatia [13].

Islamic social finance has also shown a great promise in addressing socioeconomic issues during the Covid-19 pandemic. The pandemic has had a significant impact, especially in the economic field [14]. The economic implications of Covid-19, particularly in Indonesia [15, 16], include increasing unemployment and underemployment rates, declining income levels, economic contraction, slow and uneven recovery process across sectors, including the health, economic, and education sectors. This is due to the limited sources of funding owned by the state. The government has prepared a reserve budget of USD 5.3 billion and a budget for the National Economic Recovery program of USD 37.3 billion. The funding was used for the health sector (USD 1.0 billion), social protection (USD 2.5 billion), and support for micro, small, and medium enterprises (USD 1.8 billion) [17]. The sources of funds were derived from debt, government bonds, Sukuk, and foreign debt. The Ministry of Finance [18] released the funding components of Covid-19 originating from market financing, bilateral and multilateral debts, private placement, financing from over budget balance (SAL), and the use of government endowment funds. These components certainly have limits to meet the growing needs of handling Covid-19. Consequently, the government needs to find alternative sources of funding. Islamic social finance with a great potential can be utilized to lessen the burden of funding and accelerate the recovery from the impact of the Covid-19 pandemic.

Indonesia, as a Muslim majority country, has a high potential for Islamic social finance. The potential for collecting zakat funds in Indonesia in 2020 was USD 23 billion [19], cash waqf was USD 12.6 billion per year, and the overall value of waqf land is USD 140 billion [18]. However, the collection of zakat and waqf funds is still far from its full potential. The zakat collection in Indonesia in 2019 was only USD 0.7 billion [19], while the cash waqf funds collected were only USD 0.2 billion [18]. One of the reasons for the vast gap between the realization and potential of ISF is a lack of optimal governance [20].

Furthermore, the management of ISF faces several problems, including the lack of technology utilization in its management [20], the lack of ability and competence of managers, and the lack of synergy between stakeholders [20, 21], and lack of trust and public literacy [20, 22]. Based on a survey conducted by BAZNAS (Zakat Government Institution), nationally, the level of zakat literacy is at 66.78%, which is in the middle category. Despite the country having the largest Muslim population, the literacy level for several provinces in Indonesia is still in the low category, with a score below 50%. This means that literacy is not at an advanced stage, of which this includes knowledge about the existence and functions of ISF institutions, regulations, impacts, programs, and digital payments [23].

The governance of ISF includes aspects of collection, management, distribution, and utilization of social funds. The issues of ISF governance can be categorized into six aspects consisting of Human Resources (HR), Unsupported Policies and Regulations, Technology, Sustainable Financing, Governance, and Institutional Aspect [20, 24–26]. Problems in the HR aspect include the lack of ability and motivation of human resources to develop both managers and recipients of Islamic social funds [20]. In addition, Islamic philanthropic institutions are

dominated by employees who are not from the background of Islamic Economics and Zakat Waqf management, causing the low productivity to manage ZISWAF. In addition, the low literacy related to Islamic social funds needs to be addressed by policymakers [27].

Unsupported policies and regulations become issues from the regulatory aspect [20]. First, zakat is still a voluntary act [20] and there is no sanction for those who do not carry out zakat [28]. Further, the laws that govern ISF are separated as zakat is governed by Law No. 23 of 2011, whereas waqf is governed by Law No. 41 of 2004. In addition, the infrastructure required for management needed to be addressed, particularly in the field of technology [20]. This becomes an issue that deserves consideration [29]. Many institutions do not have the required financial capacity to purchase the infrastructure needed to support technology-based governance.

Problems related to the sustainability aspects of the ISF program are of concern to IRTI [29], Pitchay et al. [24], and Sukmana [26]. Based on the findings in the field, most empowerment programs using Islamic social funds do not consider sustainability aspects and seem to end when the program ends, so that the primary goal of empowerment, namely, the transformation of *mustahiq* into *muzakki*, or *mauquf ' alaihi* into *wakif*, cannot be achieved. In addition, in the aspect of governance, Islamic social fund instruments are still partially managed. These instruments have not been integrated with the State's fiscal policy [30]. This is because Indonesia is based on "Pancasila", not a country based on Islamic law, of which its population consists of various beliefs and religions. There could be a sense of injustice if the government only regulates ands takes money from Muslims to help government finances. On the institutional aspect, there is a lack of coordination and synergy between stakeholders [20]. The lack of public trust in Islamic social finance managers [25] needs to be an essential concern for policymakers. To overcome these problems, the solution that can be done is to formulate the integration of the Islamic social finance sector and establish a sustainable management ecosystem.

The purpose of this research was to develop a strategy for the short- and long-term integration of ISF and to determine the best way to achieve that goal. This study was carried out in Indonesia, a country with a high potential for accumulating ISF. In practice, however, the ISF are only partially managed and have not been fully integrated. Raimi et al. [31], Jouti [32], and Shalleh et al. [33] conducted research on ISF integration. Previous research formulated the integration of ISF without emphasizing sustainability, and thus, the sustainability aspect in integrating Islamic social funds is highlighted in this research. Furthermore, previous studies employed a qualitative approach through case studies, literature reviews, content and thematic analysis, and a quantitative approach through Structural Equation Modeling (SEM). There has been no research that incorporates ISF that employ the two-level ANP method. The analysis of Benefit, Opportunity, Cost, and Risk (BOCR) to build the ANP framework is what makes this research unique. Several factors influenced the decision to use this method. To begin, the ANP method was used to determine the best integration in the long-term management of ISF. Second, the analysis of the benefits, opportunities, costs, and risks of each aspect was to produce findings that can be implemented in the short and long term. Third, this study identified priority solutions and strategies for achieving long-term ISF integration using the ANP BOCR method.

The findings show that the financing integration model is a priority solution in the short and long term. This demonstrates that the ISF serves several functions: Consumptive zakat is used to provide emergency assistance in the short term, productive zakat and benevolence funds are used to support economic recovery in the medium term, and waqf or sukuk are used to build long-term resilience. Furthermore, the findings of this study highlight the importance of improving the quality of HR managers through a technology-based certification and training (particularly for data analysis, e-marketing and big database management) as the best

short- and long-term strategy for implementing Islamic social finance in a sustainable manner.

The findings of this study are critical for several reasons. First, solutions and strategies for the integration of sustainable ISF are expected to solve various problems of managing different instruments of ISF, resulting in the formation of a sustainable ISF ecosystem. Second, the findings of this study serve as a resource for the ISF institutions in developing long-term strategic plans. Third, the results can provide a broader impact on people's well-being. Fourth, in the context of the Covid-19 pandemic, the findings can provide an overview of strategic policy steps and guidelines for seeking alternative funding sources by leveraging the potential of sustainable ISF.

## Literature review

### The concept of Islamic social finance and its instruments

Islamic social finance is a financial system that includes traditional and contemporary ISF instruments [32, 34–36]. Traditional ISF can be divided into two types: Philanthropy and cooperation. Zakat, *infaq*, *sadaqah*, and waqf are traditional ISF instruments based on philanthropy, while *qard* and *kafalah* are based on cooperation. Furthermore, Islamic microfinance can be categorized as contemporary ISF. Islamic social finance contributes to the third sector of the economy by assisting in creating more economic activities and reducing unemployment and poverty [37]. Several previous studies have revealed the differences in the instruments used in Islamic social finance as shown in Table 1.

Most of the previous research mentions that the instruments for ISF include zakat, *infaq*, *sadaqah*, and waqf [32, 38, 39]. Therefore, in this study, ISF instruments refer to zakat, *infaq*, al and waqf, and Islamic microfinance. Each of these instruments has different roles and functions but shares the same objective in achieving prosperity [32]. The first instrument is zakat, a mandatory economic instrument for every Muslim [40, 41]. Zakat serves as a means for distributing wealth in the society [42] from the rich to the poor to achieve socio-economic justice, increase economic growth, and reduce inequality [43]. The second instrument is *infaq* and *sadaqah*, which are the voluntary gifts by capable people to seek the pleasure of Allah [44], significantly to help the needy. Zakat, *infaq*, and *sadaqah* are economic instruments that support economic growth by increasing the purchasing power of the needy [3, 31].

As a third instrument, waqf is a social fund that plays a vital role in the Islamic framework. An asset owner donates and dedicates movable or immovable assets for the benefit of the community in perpetuity or impermanence. Waqf can be used to provide a variety of free community services, such as the construction of mosques, hospitals and healthcare centers, *madrasas*, education centers, and libraries [26]. Waqf is a vital tool in the Islamic economic system used to meet the needs of the poorest communities and promote community economic growth [26]. Furthermore, waqf, zakat, and sadaqah are instruments capable of acting as the national income and transfer payments used for redistributive purposes in an Islamic economy [6].

**Table 1. Instruments of Islamic social finance.**

| No | Researchers | Year | Zakat | *Infaq* | *Sadaqah* | Waqf | *Qard* | *Qard* Hasan | *Kafalah* | Gift | Islamic Microfinance |
|----|-------------|------|-------|---------|-----------|------|--------|--------------|-----------|------|----------------------|
| 1. | Jouti | 2019 | ✓ | | ✓ | ✓ | ✓ | | ✓ | | ✓ |
| 2. | Usman et al. | 2019 | ✓ | ✓ | ✓ | | | | | | |
| 3. | Adjar et al. | 2020 | ✓ | | | ✓ | | ✓ | | | |
| 4. | Hudaefi | 2020 | | ✓ | ✓ | ✓ | | | | | |

Islamic microfinance is the fourth instrument of ISF. Islamic microfinance serves two purposes: Social and commercial. As a social institution, it manages zakat, *infaq*, *sadaqah*, and waqf in accordance with the Syariah. On the other hand, as a commercial institution, it conducts businesses and provides financing for economic development. Islamic microfinance focuses on economic development through capital assistance and the empowerment of the marginal community with no access to capital to improve their well-being [45].

## Sustainability of Islamic social finance

Sustainability refers to the capacity to operate over a long period and continuously [46]. Sustainable growth has become a global development goal in both government and non-governmental organizations [47]. There are two paradigms in sustainability, namely, the institutionalist and welfarist [48]. The institutionalist paradigm considers that financial self-sufficiency is the key to achieving sustainability. Thus, a person can be said to be sustainable if he or she achieves financial independence [48]. In contrast, the welfarist paradigm considers that social functions in poverty alleviation are more important in achieving sustainability. Both paradigms are essential in the context of ISF. Financial self-sufficiency refers to an institution's ability to fund its operations independently, while also benefiting others. On the other hand, the main goals of ISF are poverty alleviation and its reachability [49–51]. Three factors contribute to the sustainability of Islamic social finance. First, the collection's sustainability is linked to the fact that funds will always be collected from the community. To ensure the long-term viability of funds supply, all information about zakat and its administration must be transparent and easily accessible to the public [52]. Second, institutional sustainability is linked to the institution's ability and availability to carry out operational tasks [48]. Third, the sustainability of distribution is linked to how the Islamic social finance institution performs its social and intermediary functions to improve welfare and achieve broader benefits [50].

In practice, the empowerment programs of ISF institutions have not been able to transform the poor into prosperous. Furthermore, many of these programs come to an end after a certain period and do not track whether the poor have truly become self-sufficient and prosperous. Several factors contribute to the unsustainable nature of ISF institutions' empowerment projects [20]. To begin, massive sums of money are required to provide effective and optimal empowerment. Second, qualified human resources with leadership qualities are needed [53]. Third, the beneficiary's desire and ability to develop and change into *muzakki* or *munfiq* are required [3, 54].

## Issues with Islamic social finance integration

In order to achieve optimal management, ISF must overcome several challenges. A lack of motivation among beneficiaries to develop, a lack of synergy and coordination among stakeholders, a lack of human resources capabilities and quantity to manages Islamic social finance, a lack of ability to optimize technology, regulations that are still voluntary, and a lack of awareness and trust of *muzakki* in Islamic social finance institutions are among the problems encountered by Islamic social finance [20, 24–26, 55]. Further, the ISF instruments are still partially (separatedly) managed, which is demonstrated by several points. First, according to the 2017–2022 Indonesian Zakat Architecture, zakat management is supported by regulations but remains a voluntary instrument, not an obligatory instrument [20]. Second, ISF instruments are governed by different regulations and laws. The management of zakat is regulated in Law no. 23 of 2011, while the management of waqf is regulated in Law no. 41 of 2004. Further, ISF instruments are managed by various institutions (government, corporate and community organizations) [56] of which there exist differences in the rights, obligations, and

interests of different institutions. Fourth, the regulators of zakat and waqf are not the same. The National *Amil* Zakat Agency (BAZNAS) regulates zakat, while the Indonesian Waqf Board (BWI) is the waqf management regulator. However, BAZNAS and BWI are eventually accountable to the Minister of Religion Affairs. Based on these challenges, the integration of ISF must be carried out.

## Integration model of Islamic social finance

Integration refers to the process of blending into a single unit, thus, integration in ISF combines funds management, institutions, governance, functions, etc. According to Balassa [57], integration offers various advantages. First, integration allows for the mobility of production elements, which has a positive impact on enhancing welfare. Second, integration reduces costs, resulting in an increase in output. Third, integration promotes job specialization. Fourth, integration creates a welfare through policy harmonization. Fifth, integration can increase technology adoption. Furthermore, integration has a favorable impact on the socio-economic aspects of the community, including: (1) Improving sustainability performance [58], (2) enhancing economic growth [59], and (3) improving socio-economic fairness [2].

Raimi et al. [31] integrated corporate social responsibility, zakat, and waqf into the faith-based model to reduce poverty, promote business development, and strengthen the economy of Muslim-majority communities. Hassanain [60] conducted a literature review on microfinance, waqf, and zakat to develop an integration model. Haneef et al. [45] used the Structural Equation Model (SEM) analysis to examine the significant influence between differencct aspects of Islamic social finance that include waqf, takaful, human resource management, and financing. Pitchay et al. [24] conducted a literature review to develop a cooperative-waqf model. Further, using the thematic analysis, Ambrose et al. [61] developed a waqf financing model for public goods and mixed public goods in Malaysia, where the proceeds of waqf investment can be used to fund 11 items of government spending.

A research that specifically discusses the sustainability of the ISF model was carried out by Jouti [32], which focuses on creating an ISF ecosystem, and by Shalleh et al. [33] which focuses on the integration of waqf-based takaful for flood victims in Malaysia. This study developed the previous research by building an integration model between ISF instruments with an emphasis on sustainability aspects using the two-level ANP method.

## Method

### Research method and analysis approach

The Analytical Network Process (ANP) method was used in this qualitative study, along with the analysis of Benefit, Opportunity, Cost, and Risk (BOCR). This two-level ANP method has the characteristics of mathematically examined conditions and inputs obtained based on several constructs, allowing it to aid the decision-making from various options and complex data [62]. The ANP method approach was chosen because ISF include various instruments, each with its own set of characteristics and issues, both tangible and intangible. In this study, the ANP approach was the best choice for producing strategic decisions [63]. Meanwhile, the BOCR analysis was selected for a variety of reasons, to begin, every decision has both positive and negative consequences, which must be considered when determining priority decisions. Second, the BOCR approach was used to make decisions that can be implemented in the short and long terms. The ANP BOCR method prioritizes solutions and strategies to develop a sustainable ISF integration model. The ANP method of research has three phases, which are depicted in Fig 1 [64].

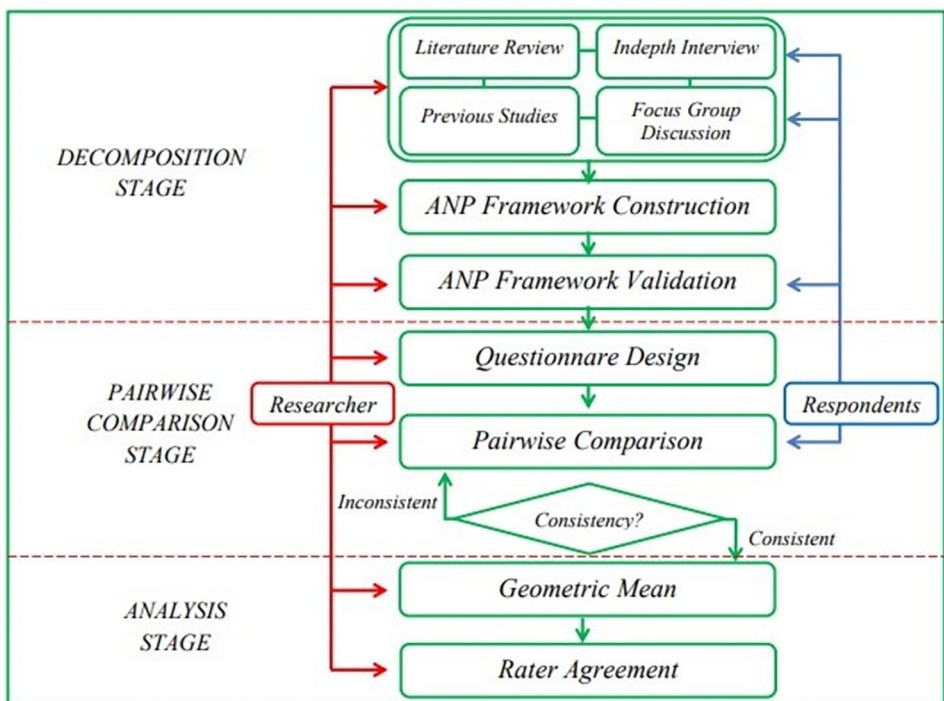

**Fig 1. ANP BOCR research stage.**

First, the decomposition phase sought to construct and validate the ANP model through focus group discussions and in-depth interviews with experts in ISF among academics, practitioners, associations, regulators, and community leaders. A literature review was conducted as part of this study to strengthen the ANP model that was developed. A literature review analysis was used to collect problems, solutions, and strategies discovered and analyzed by previous researchers to integrate sustainable ISF. In addition, the literature review analysis was used to validate the findings from the focus groups discussion and in-depth interviews.

Three Focus Group Discussions (FGDs) were held. The first and second FGDs were held with experts to identify aspects, solutions, and strategies for integrating sustainable ISF and identify benefits, opportunities, costs, and risks to determine the short-term and long-term analyses. The model developed through FGD 1 and 2 was then strengthened through in-depth interviews and literature reviews. Following that, the third FGD validated the constructed ANP model. The second was the pairwise comparison phase to design a questionnaire in the form of a pairwise comparison to determine and analyze aspects, solutions, and priority strategies in integrating sustainable ISF. The questionnaire that the respondents had answered were used for the next phase. The third was the data analysis phase, which aimed to calculate the priority value of each aspect, solution, and strategy formulated using the Microsoft Excel and Super Decision software. The use of the ANP method with the BOCR approach made it possible to produce realistic and net values. The net value/additive negative describes alternative options that are more profitable or ideal to be prioritized in the long term, while realistic/multiplicative value represents more favorable outcomes for short-term decisions [63, 65]. According to Saaty and Vargas [63] there are several steps to synthesize each priority option from each criterion where:

$$\text{Net Value/additive negative} = bB + oO - cC - rR \qquad (1)$$

$$\text{Realistic/multiplicative} = (BxO)/(CxR) \qquad (2)$$

B or b stands for Benefit, O or o stand for Opportunity, C or c stands for cost, R or r stands for Risk. After the analysis phase, the next step was to validate the results with various ISF experts to strengthen the integration model that will be built.

## Data and research samples

The data used were primary data obtained through focus groups, in-depth interviews, and ANP questionnaires filled out by the respondents. The data sources gathered were from 13 experts among academics, practitioners, associations, regulators, and community leaders. In ANP research, the number of respondents is not a critical requirement for research validation [66] but their expertise is. Each expert completed the questionnaire consciously and in a mentally healthy state. The participants were required to fill in a specific consent form beforehand to ensure that they were not subject to any pressure from any party to participate in the interviews. The research team guaranteed the confidentiality of the experts' data and responses and is fully responsible for all applicable laws. Applicable laws will account for the disclosure of confidential data and experts' responses.

In terms of research ethics, this research followed the rules and ethics of research conduct and writing coordinated by the Institute for Research and Community Service (LPPM), Universitas Airlangga and those mandated by law. It is also explained in the Airlangga University Chancellor's Regulation Number 34 of 2019 concerning the Rules of Conduct clause 16 (b) that researchers must be honest, objective, and adhere to all aspects of the research process and may not falsify or manipulate data or research results, as well as clause 16 (f) that researchers must respect and appreciate the object of research, whether in the form of humans or animals, both living and dead, or parts/fragments of the object of research. In addition, this research was supervised by the Center for Research and Publications (3P) within the Faculty of Economics and Business Universitas Airlangga, which functions as an ethics committee according to the Dean Decree Number 88/UN3.1.4/2020.

The purposive sampling technique aims to select respondents according to the criteria required in the ANP method. For this study, the selected respondents must have the following criteria: (1) Respondents are experts in the field of ISF among academics, practitioners, associations, regulators, and community leaders, (2) have more than two journal publications in the field of ISF for academic representatives, and (3) for representatives of regulator, practitioners, and associations, they should hold an important position and bear a great responsibility in the management of ISF in their respective institutions. The research team contacted the prospective (representative) respondents by phone and email before conducting the interviews to inquire about their willingness to complete the questionnaire. There were two options for completing the questionnaire: Online or offline. For the respondents who opted to complete the questionnaire offline, the research team attended to them, while adhering to a strict health protocol, by still maintaining a safe distance and wearing a mask. For respondents who opted to fill out questionnaires via teleconference zoom, they were accompanied by three team members with each job desk either as a moderator, a note-taker, or a backup. The list of respondents in this study is shown in Table 2.

The questionnaire of the study is in the form of a pairwise comparison of the benefits, opportunities, costs, and risks of related aspects in integrating sustainable ISF. Governance, sustainable financing, institutions, regulations, technology, and human resources are among the issues at stake in integrating the instruments of ISF. The questionnaire also comprises

**Table 2. List of research respondents.**

| No | Representative | Name | Institution | Affiliation |
|----|----------------|------|-------------|-------------|
| 1 | Association | KH | National Zakat Forum in East Java | Leader |
| 2 | | BS | National Zakat Forum | Leader |
| 3 | Practitioner | CW | Zakat Institution "*Manajemen Infak*" | Director of mobilization, research, and transformation of Zakat Institution |
| 4 | | MHZ | BAZNAS Centre Strategic of Studies | Director |
| 5 | Regulator | SPY | Ministry of Religious Affairs | Staff at Ministry of Religious Affairs |
| 6 | | SEH | National Committee on Islamic Economy and Finance (KNEKS) | Director of Sharia Ecosystem Infrastructure |
| 7 | | AJ | National Committee on Islamic Economy and Finance (KNEKS) | Director of Sharia Social Finance |
| 8 | Academics | NH | Universitas Yarsi | Professor in Islamic Economics |
| 9 | | RS | Universitas Airlangga | Professor in Islamic Economics |
| 10 | | MSN | Universitas Indonesia | Head of Sharia Finance Education and Human Resources Development Division |
| 11 | Community Leader | MTH | Department of Population and Civil Registration | Secretary |
| 12 | | MZK | Al Mustofa Educational and Social Foundation | Founder |
| 13 | | NS | Indonesian Zakat Initiative (IZI) | Program Director |

various solutions and strategies to achieve the research objectives. The pairwise comparison assessment in the questionnaire is rated on a scale of 1–9, where 1 (one) means not important/needed/influential and 9 (nine) means absolutely important/needed/influential.

## Result

This study aimed to integrate sustainable ISF by building solutions and priority strategies in the short and long term. Before developing solutions and strategies, identification of important aspects was carried out to provide a comprehensive analysis of the most appropriate priority solutions and strategies in integrating sustainable ISF.

### Priority aspect

Human resources, regulation, technology, sustainable financing, governance, and institutions are six aspects related to the integration of sustainable ISF, according to the FGD and in-depth interviews. Table 3 shows the aspects with the highest priority based on the ANP data processing.

The data processing results show that human resource is the highest priority aspect in ISF institutions, with a value of 0.97181. Human resource refers to all people/staff/employees who participate in the management of ISF [20]. With a value of 0.8689, the regulation comes in as the second priority. Regulations are all rules put in place to provide legal guidance and support

**Table 3. ANP BOCR result of priority aspect.**

| Aspect | Code | Benefit | Opportunity | Cost | Risk | Realistic | Rank |
|--------|------|---------|-------------|------|------|-----------|------|
| Human Resources at Islamic Social Finance Institution | A | 0.1029 | 0.19763 | 0.16966 | 0.12334 | 0.97181 | 1 |
| Regulation | B | 0.15479 | 0.09765 | 0.11073 | 0.1571 | 0.8689 | 2 |
| Supporting Technology | C | 0.24925 | 0.16373 | 0.24824 | 0.214 | 0.7682 | 3 |
| Sustainable Financing | D | 0.1229 | 0.06846 | 0.10973 | 0.10029 | 0.7645 | 4 |
| Governance | E | 0.23496 | 0.33413 | 0.17927 | 0.25364 | 0.7152 | 5 |
| Institution | F | 0.13521 | 0.13841 | 0.18238 | 0.15163 | 0.6767 | 6 |

for the management and development of ISF [67]. With a value of 0.7682, the third priority aspect is the use of technology, while sustainable financing comes in the following priority order with a value of 0.7645. Sustainable funding is associated with business development and achieving financial independence using Islamic social funds [68]. Governance is the fifth priority aspect with a value of 0.7152. The collection, distribution, and utilization process in ISF are governed by transparency, accountability, responsibility, independence, and fairness [69]. The institutional aspect comes as the least priority, with a score of 0.6767. The institutional aspect relates to the structure and coordination system. In Indonesia, ISF institutions can take the form of government and private legal entities [70]. For zakat management, the legal entity can take in the form of Amil Zakat Institution (LAZ) and the National Amil Zakat Agency (BAZ-NAS), while Nazir is the legal entity for the management of waqf.

## Priority solution in integrating sustainable Islamic social finance

There are eight solutions to integrate sustainable ISF based on the findings of the focus group, in-depth interviews, and literature review which include:

SL.1 *Integrating ISF with state fiscal instruments, which is supported by existing regulations.* Integrating Islamic social finance with the fiscal policy can be accomplished by making zakat a tax credit or tax deduction.

SL.2 *Integration through non-formal coordination between ISF institutions.* The Islamic social finance institutions may establish joint coordinations to conduct empowerment programs, while prioritizing the work specialization of each institution.

SL.3 *Integration among Islamic social finance institutions based on capacity building and role division.* This type of integration includes vertical and horizontal integrations [70]. The vertical integration is achieved through a collaboration between ISF institutions and the government. The government collaborates by providing a database of people who are eligible to receive Islamic social finance funds. The government also provides rules and guidance for the management of ISF. Meanwhile, a horizontal integration is conducted between ISF management institutions, where different institutions collaborate to carry out joint social programs [32]. In addition, a horizontal integration is also carried out with academics and associations for future management development of the ISF [20]. Some organizations concentrate on the health sector, while others concentrate on economic empowerment and education. As a result, combining institutions based on their specialization may be the best option for achieving the research objective.

SL.4 *Establishing an integrated national data center for all ISF instrument.* The data that must be integrated include financial statements, donor funds, beneficiary data, and other relevant information.

SL.5 *Integration based on regional clusters with a solid authority.* This integration may concentrate on integrating Islamic social finance institutions into geographical groups.

SL.6 *Integrating ISF through the passage of new laws and regulations.* Each instrument of ISF, currently, has its own set of rules and regulations

SL.7 *Integration based on ISF management clusters*, such as zakat *Amils* joining other zakat institutions and waqf manager joining other waqf institutions

Table 4 shows the ANP result of solution and the order of priority solutions in the short term, long term, and based on experts' agreement.

**Table 4. ANP results of solution.**

| Cluster | Code | Benefit | Opportunity | Cost | Risk | Net Value | Net Value Priority | Realistic | Reaslistic Priority | Expert Judment | Expert Judgment Priority |
|---------|------|---------|-------------|------|------|-----------|---------------------|-----------|---------------------|----------------|--------------------------|
| Solution | SL.1 | 0.0526 | 0.0588 | 0.0466 | 0.04593 | 0.0188 | 1 | 1.4435 | 1 | 0.2118 | 2 |
| | SL.2 | 0.0277 | 0.0262 | 0.0174 | 0.02957 | 0.0070 | 2 | 1.4149 | 2 | 0.1016 | 4 |
| | SL.3 | 0.0258 | 0.0235 | 0.0227 | 0.02338 | 0.0032 | 5 | 1.1452 | 3 | 0.0952 | 5 |
| | SL.4 | 0.0297 | 0.0204 | 0.0226 | 0.023715 | 0.0038 | 4 | 1.1313 | 4 | 0.0929 | 6 |
| | SL.5 | 0.0555 | 0.0657 | 0.0537 | 0.063225 | 0.0042 | 3 | 1.0737 | 5 | 0.2438 | 1 |
| | SL.6 | 0.0244 | 0.0144 | 0.0260 | 0.020225 | -0.0073 | 7 | 0.6719 | 6 | 0.0787 | 7 |
| | SL.7 | 0.0159 | 0.0289 | 0.0380 | 0.020995 | -0.0141 | 7 | 0.5777 | 7 | 0.1071 | 3 |
| | SL.8 | 0.0180 | 0.0118 | 0.0226 | 0.022965 | -0.0157 | 8 | 0.4113 | 8 | 0.0685 | 8 |

The formulated solution was then processed using the ANP BOCR to determine the priority order of solutions. Short term solution considers the marginal cost of the benefits obtained. On the contrary, in the long term, the priority is achieved when the total number of benefits and opportunities are obtained after reducing the total cost and risk [63, 65]. On the other hand, an expert judgment does not consider the benefits, opportunity, cost, and risk aspects of each solution formulated in this study. Table 5 summarizes the results of data processing.

According to the data processing results, the first and second priorities in the short and long-term are in the same order. The solutions that will be implemented first and foremost in the short and long term are financial integration in providing funding to beneficiaries and integration of Islamic social finance with the state fiscal policy. According to expert judgment, integrating national data centers is the highest priority solution, with financial integration in providing funding to beneficiaries coming in second.

## Priority strategies in integrating sustainable Islamic social finance

There are nine strategies to integrate sustainable ISF based on the findings of the FGD, in-depth interviews, and literature review. The first strategy (ST.1) is to improve the quality of human resources through training and certification. The second strategy (ST.2) promotes the availability of real-time data through database integration and reporting. The third strategy (ST.3) is optimizing the digital technology in Islamic social finance instruments collection, management, and distribution process. The fourth (ST.4) is a coordination and synergy among stakeholders (e.g., practitioners, regulators, associations, and academicians) to establish shared commitment and vision. The fifth (ST.5) is encouraging the broadening of the range of benefits by establishing service units at lower levels, such as service units at the district, village, and mosque levels, as well as strategic locations that support the broadening of the reach of the benefits of ISF instruments. The sixth (ST.6) is establishing a new institution that is capable of integrating all instruments of ISF. In Indonesia, various institutions manage different ISF

**Table 5. Summary of priority solutions.**

| Solution/Rank | Solution in Sustainable Islamic Social Finance Integration | | |
|---------------|---------------------------------------------|---|---|
| | 1 | 2 | 3 |
| Expert Judgment | Integration of national data centers (SL.5) | Financial integration in providing funding to beneficiaries (SL.1) | Integration through the establishmen of new laws and regulations (SL.7) |
| Short Term Realistic Value | Financial integration in providing funding to beneficiaries (SL.1) | Integration with state fiscal policy (SL.2) | Integration through non-formal coordination between institutions (SL.3) |
| Long Term Net Value | Financial integration in providing funding to beneficiaries (SL.1) | Integration with state fiscal policy (SL.2) | Integration of national data centers (SL.5) |

**Table 6. ANP Results of strategies.**

| Cluster | Code | Benefit | Opportunity | Cost | Risk | Net Value | Net Value Priority | Realistic | Reaslistic Priority | Expert Judgment | Expert Judgment Priority |
|---|---|---|---|---|---|---|---|---|---|---|---|
| Strategies | ST.1 | 0.0455 | 0.0393 | 0.0377 | 0.030375 | 0.0168 | 1 | 1.5651 | 1 | 0.1546 | 3 |
| | ST.2 | 0.0334 | 0.0437 | 0.0297 | 0.03745 | 0.0099 | 2 | 1.3113 | 2 | 0.1515 | 4 |
| | ST.3 | 0.0500 | 0.0463 | 0.0444 | 0.048875 | 0.0030 | 3 | 1.0676 | 3 | 0.1885 | 1 |
| | ST.4 | 0.0364 | 0.0450 | 0.0443 | 0.03603 | 0.0011 | 4 | 1.0289 | 4 | 0.1666 | 2 |
| | ST.5 | 0.0235 | 0.0148 | 0.0181 | 0.02406 | -0.0038 | 6 | 0.8001 | 5 | 0.0752 | 6 |
| | ST.6 | 0.0086 | 0.0087 | 0.0089 | 0.01088 | -0.0023 | 5 | 0.7807 | 6 | 0.0365 | 9 |
| | ST.7 | 0.0173 | 0.0179 | 0.0213 | 0.019505 | -0.0055 | 7 | 0.7485 | 7 | 0.0750 | 7 |
| | ST.8 | 0.0214 | 0.0140 | 0.0203 | 0.02178 | -0.0066 | 8 | 0.6796 | 8 | 0.0723 | 8 |
| | ST.9 | 0.0135 | 0.0199 | 0.0250 | 0.02105 | -0.0126 | 9 | 0.5124 | 9 | 0.0795 | 5 |

instruments. The seventh strategy (ST.7) is realizing the government's role in supporting the integration of ISF through regulation and socialization; several previous studies have concluded that government support for integrating ISF is still inadequate. The eighth strategy (ST.8) focuses on developing the ISF institutions to have two functions, namely, *Amil* and *Nazir*. Several Baitul Maals wat Tamwiils (BMT) have operated the two functions under the same body. The ninth strategy (ST.9) is revising and updating all laws and regulations governing the management of Islamic social finance instruments to accommodate the development of ISF instruments. This development includes the management of zakat on profession, corporate zakat, temporary waqf, stock waqf, etc. The ANP results and order of priority strategies is shown in Table 6, which displays the short-term, long-term priority solutions, and expert judgment.

After developing the solution, a strategy must be developed. Strategies are employed to aid in the achievement of solutions, and thus, the integration of ISF will be realized. The strategies are summarized in Table 7.

The results show that the short and long-term strategies are in the same order. Improving the quality of human resources, establishing real-time database integration and reporting, and optimizing the use of digital technology, become the priority strategies. However, the expert judgment considers the optimization use of digital technology to be the priority strategy.

## Discussion

### Priority solution in developing a sustainable Islamic social finance integration model

Financial integration in providing financing to beneficiaries is a high-priority solution that can be implemented in the short and long term. Further, this solution is very consistent with the

**Table 7. Summary of priority strategies.**

| Strategies | Strategies in Sustainable Islamic Social Finance Integration | | |
|---|---|---|---|
| | 1 | 2 | 3 |
| Expert Judgment | Optimizing the use of digital technology (ST.3) | Coordination and synergy between stakeholders (ST.4) | Improving the quality of human resources (ST.1) |
| Short Term Realistic Value | Improving the quality of human resources (ST.1) | Establishing real time information through database integration and reporting (ST.2) | Optimizing the use of digital technology (ST.3) |
| Long Term Net Value | Improving the quality of human resources (ST.1) | Establishing real time information through database integration and reporting (ST.2) | Optimizing the use of digital technology (ST.3) |

sustainability paradigm, where the funding provided will broaden the range of benefits, and ultimately creates financial independence for the beneficiaries, as stated by Haneef et al. [45], Azganin et al. [71], Thaker et al. [72], and Widiastuti et al. [20]. Sari et al. [73] show that by distributing zakat, the time taken by the poor to exit poverty is faster (3.3 years) than without zakat (6.6 years). In addition, a study conducted by Widiastuti et al. [3] analyzed financing programs carried out by the ISF institutions, including Yatim Mandiri, Dompet Dhuafa, Inisiatif Zakat Indonesia, Nurul Hayat, YDSF, and Dompet Amanah Umat, and found that they have a significant impact on increasing the welfare of the poor. The distribution aspect, particularly productive distribution, holds the key to success in ISF integration in transforming the poor into the capables [69]. A financial integration through ISF instruments, such as zakat and waqf, will increase production factors. In this case, financing will be more effective due to an increase in the amount of funds obtained by the poor to achieve welfare. This is consistent with the economic integration theory discussed by Balassa [57]. This also supports Sanyinna and Omar [74] claim that Islamic financing benefits small businesses by maximizing Islamic social financial instruments (zakat, *infaq*, *sadaqah*, and waqf) as the best strategy for improving the recipients' quality of life. The financing provided through integrating Islamic social finance instruments has a cheap cost of the fund, and thus, it ensures the beneficiaries' welfare. Furthermore, the integration of financing demonstrates that Islamic social fund instruments serve multiple purposes: Consumptive zakat is used to provide an emergency assistance in the short term, while productive zakat and benevolence funds are used to support economic recovery in the medium term, and waqf or Sukuk are used to build long-term resilience. The financial integration of Islamic social finance instruments has also occurred in several institutions, although only on a micro-scale. Several institutions in Indonesia have worked to integrate zakat and waqf funds. Dompet Dhuafa, for example, integrated zakat and waqf funds when constructing an Integrated Health Home (Hospital). Cash Waqfs are used to build infrastructure on waqf land, whereas zakat funds provide free health care to the poor. Further, recently, Dompet Dhuafa, BAZNAS, and BWI have collaborated to integrate zakat and waqf, the newly established MoU, to collaborate on social projects.

This priority, however, differs from expert opinion, which emphasizes the importance of integrating nationally centralized data, such as database on donors, beneficiaries, and financial reports. According to Afriadi and Sanrego [54] and Hudaefi et al. [75] the formation of national-centered data is critical for optimizing the integration and management of ISF. The integration of beneficiary-related information is crucial for detecting data on beneficiary development and preventing overlapping distributions. Since ISF are still managed in a fragmented manner (partially), no single platform currently integrates the data from all instruments of ISF in Indonesia. As the latest development, a data integration platform has been established in the zakat sector. BAZNAS, an Indonesian zakat institution and regulator, released SIMBA (BAZNAS Information Management System), an integrated platform for zakat database management from all BAZNAS and LAZNAS across Indonesia. This platform only integrates the data from national zakat collections. As a result, establishing a central platform capable of integrating data on all Islamic social financial instruments is critical.

Financial integration and integration of national data centers are interrelated. This is so since funding distribution necessitates specific data to ensure the long-term viability of the funds provided. Data integration is needed to reduce the potential risk and realize the collection of distributed financings so that the supply of funds will be guaranteed. Further, the integration of beneficiary data is needed to ensure the depth of the reach of benefits and the creation of financial independence, and thus, sustainability will be achieved. On another side, financial integration remains a short- and long-term priority because it is directly related to the primary goal of Islamic social finance, which is to create prosperity by improving the

business and economic capacity of beneficiaries. As one of the priority solutions in this research, the regulator must provide a legal environment to accommodate all changes and developments in the management of Islamic social finance. First, the government must revise the law to accommodate the rapid growth of Islamic social finance instruments [76]. Second, the government must socialize the revised laws and regulations to all stakeholders, and make the law more socially acceptable [20, 77]. Third, encourage ISF management institutions to implement the law by offering rewards or incentives that the Institution can use, such as free training and accompaniment [20].

## Strategic priorities in developing a sustainable Islamic social finance integration model

According to expert judgments, the strategy with the highest priority is optimizing the use of digital technology. This in line with Thaker et al. [78], Usman et al. [79], Berakon et al. [80], Nor et al. [81], and Widiastuti et al. [20], all of which emphasize the importance of using technology in managing the instrument of ISF. First, technology for creating a centralized database linked to the government and ISF supervisors is one example of a technology that can be used [20]. This strategy is also linked to the immediate solution, which is the consolidation of national data centers for all Islamic social and financial instruments. To become effective and efficient, data integration necessitates the creation of a sophisticated web-based platform that all stakeholders can access. Second, technological advancements in supervision and monitoring [20]. Third, technology in the marketing component to aid in the collection of target achievement [82, 83]. Fourth, technology to generate real-time management reports and make them available to the public [20].

However, in terms of short-term and long-term priorities, optimizing the use of technology has slipped to third place. Experts consider improving the quality of human resources to be the most important aspect, both in the short and long term. The importance of enhancing the capability of human resources goes in line with Karami et al. [84], Hidayatullah and Priantina [21], Ali et al. [55], and Widiastuti et al. [20]. The difference in priorities between expert judgments and short/long term orientations occurs because of technology investment, which is considered a long-term investment, necessitates a significant funding source, which some zakat institutions may not afford [20]. Optimizing the use of technology, which requires a large amount of investment, is a challenge for Islamic social finance institutions, particularly in Indonesia. This is due to the fact that the operational costs incurred by ISF institutions are determined by the amount of funds that can be raised. Meanwhile, there is a significant gap between the realization and potential of ISF. As a result, institutions prefer to improve competencies and capabilities of human resources, relevant to their respective fields of work compared to improving technological features. This is also consistent with the aspects of BOCR addressed in this study. The cost of optimizing technology use is 0.0537, higher than the cost of improving the quality of human resources, which is around 0.0466. Costs associated with technology investments include acquiring digital channels, website management, social media, purchase of computers, and maintenance cost.

Furthermore, when determining priority strategies in the short and long term, experts also consider the risks exposed. The risk of optimizing the use of technology is high (0.0632), with the benefits obtained are also higher (0.0555). Meanwhile, the threat from the institution's strategy to improve the quality of human resources is lower at 0.0459. Cybercrime is one of the risks of optimizing the use of technology in ISF [85]. To mitigate this risk, competent human resources in their field are required. Implementing technology without qualified human resources will result in sub-optimal management. This is consistent with the findings of

Widiastuti et al. [20] that a limited number of human resources is capable of optimizing the use of advanced technology, particularly in those newly established institutions. Therefore, experts place qualified human resources as the prioritized strategy to mitigate risks, not only risks associated with technology application, but all risks associated with the integration of sustainable Islamic social finance.

Optimizing the use of technology and improving human resource quality can be done concurrently as stated by Widiastuti et al. [20] that highlights the importance of improving the quality of zakat management resources, particularly in the technological aspect, by emphasizing technology intensification (by developing *Amils'* ability to use technology) and extensification (by increasing the number of *Amils* who master technology). Improving the quality of human resources is the foundation for the long-term integration of ISF, while technology facilitates an optimal ISF management.

Finally, in order to integrate sustainable ISF, stakeholders must prioritize solutions and strategies, by considering the benefits, opportunities, costs, and risks (BOCR) of each priority solution and strategy. Improving the quality of human resources and optimizing the use of digital technology are two strategies that must be implemented immediately. Based on the short- and long-term analysis and expert consensus, the two strategies are given the highest priority.

## Managerial relevancy

This study has several implications for stakeholders. The findings of this study emphasize solutions and priority strategies for integrating ISF, with Islamic social finance institutions playing a critical role in providing certification and training to improve the quality of human resources. Each training activity and accreditation must be given in accordance with the employee's field of work. Technology optimization, such as big data management, data security, e-marketing, and others, are examples of certification and training that must be provided. Human resource quality must be directed to obtain management and technical skills. Managers are encouraged to have managerial skills, such as making strategic decisions, forecasting, and imposing rewards and punishments. Staff are encouraged to learn technical skills, such as collection program operation, *mustahik* guidance, reporting, and administration.

The government, as a regulator, must assist Islamic social finance institutions in providing facilities or infrastructure to improve the quality of human resource managers, such as free training and certification for managers. Further, no law or regulation has required ISF managers to have standardized or specific competencies. As a result, one form of input for the revision of the law and regulation requires all ISF managers to have specific competencies in managing Islamic social finance institutions so that these institutions can be managed more professionally.

This study also emphasizes the significance of financial integration and data integration on the national scale. Associations involved in Islamic social finance can use the research findings to establish strategic planning on integrating ISF instruments. From an academic standpoint, the results of this study develop the previous studies on the integration of the ISF model, which were conducted without using specific mathematical methods. This study used a systematic approach to create an ideal integration model that can be of priorities both in the short and long term, while also taking the aspect of BOCR into account.

## Conclusion

This research used the ANP BOCR to develop priority solutions and strategies to integrate Islamic social finance based on expert agreement, short-term and long-term analysis. The aspect of human resources, regulation, and technology use become the highest aspect to be

considered in achieving the research objective. Priority solutions in the short and long term have the same order of priorities. Financial integration in providing funding to beneficiaries and integration with the state fiscal policy is the first and second priority solution. Meanwhile, integration through a non-formal coordination between institutions is the third priority solution in the long term. Besides, integrating national data centers has been the third priority solution for a long term and the highest solution based on expert judgments. In essence, the resulting priority solutions relate to priority aspects include financing, regulation, and technology.

As a strategy, improving the quality of human resources, establishing real-time information through database integration and reporting, and optimizing digital technology are three priority strategies that can be implemented in the short and long term. Furthermore, the experts mentioned the need for coordination and synergy between stakeholders in integrating sustainable Islamic social finance. To successfully incorporate Islamic social finance and ensure sustainability, stakeholders are encouraged to implement the research results (solutions and strategies).

## Limitation and future research

This research only focuses on six aspects to develop ideal solutions and strategies, however, this research has thoroughly examined each aspect's benefits, opportunities, costs, and risks. Future research can consider essential elements that have not been addressed in developing a sustainable Islamic social finance integration model, such as cultural aspects, donor behavior, and government support.

## Supporting information

**S1 Appendix. Questionnaire.**
(DOCX)

## Acknowledgments

The authors would like to thank Center Research and Publication (3P) of Faculty Economic and Business Universitas Airlangga as ethics committee. The authors would also like to thank all academicians, regulators, associations, practitioners, and community leaders for their support in this research. The authors also appreciate the research team members for their support in the completion of this research.

## Author Contributions

**Conceptualization:** Tika Widiastuti, Anidah Robani, Sri Herianingrum.

**Data curation:** Puji Sucia Sukmaningrum.

**Formal analysis:** Puji Sucia Sukmaningrum, Imron Mawardi, Sri Ningsih, Sri Herianingrum.

**Funding acquisition:** Tika Widiastuti.

**Investigation:** Imron Mawardi, Sri Ningsih, Sri Herianingrum, Muhammad Ubaidillah Al-Mustofa.

**Methodology:** Puji Sucia Sukmaningrum, Imron Mawardi, Sri Ningsih, Sri Herianingrum, Muhammad Ubaidillah Al-Mustofa.

**Supervision:** Anidah Robani.

**Writing – original draft:** Muhammad Ubaidillah Al-Mustofa.

**Writing – review & editing:** Anidah Robani.

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
