## [Decision Letter · Decision Letter 0]

9 Dec 2021

PONE-D-21-31476Integrating Sustainable Islamic Social Finance: Analytical Network Process using Benefit Opportunity Cost Risk (ANP BOCR) Framework: The Case of IndonesiaPLOS ONE

Dear Dr. Robani,

Thank you for submitting your manuscript to PLOS ONE. After careful consideration, we feel that it has merit but does not fully meet PLOS ONE’s publication criteria as it currently stands. Therefore, we invite you to submit a revised version of the manuscript that addresses the points raised during the review process.

We look forward to receiving your revised manuscript.

Kind regards,

Elisa Ughetto

Academic Editor

PLOS ONE

“This research grant is a Covid-19 Research Mandate with the contract number 1025/UN3.15/PT/2021. The authors would like to thank the Research and Community Service Institution (LPPM) of Universitas Airlangga for their financial assistance. The authors would also like to thank all academicians, regulators, associations, practitioners, and community leaders for their support in writing this research.”

“This research grant is a Covid-19 Research Mandate with the contract number 1025/UN3.15/PT/2021. The authors would like to thank the Research and Community Service Institution (LPPM) of Universitas Airlangga for their financial assistance. The authors would like to thank Center Research and Publication (3P) of Faculty Economic and Business Universitas Airlangga as ethics committee. The authors would also like to thank all academicians, regulators, associations, practitioners, and community leaders for their support in writing this research.”

 “This research grant is a Covid-19 Research Mandate with the contract number 1025/UN3.15/PT/2021. The authors would like to thank the Research and Community Service Institution (LPPM) of Universitas Airlangga for their financial assistance. The authors would also like to thank all academicians, regulators, associations, practitioners, and community leaders for their support in writing this research.”

“No authors have competing interests”

6. We note that you have indicated that data from this study are available upon request. PLOS only allows data to be available upon request if there are legal or ethical restrictions on sharing data publicly. For more information on unacceptable data access restrictions, please see http://journals.plos.org/plosone/s/data-availability#loc-unacceptable-data-access-restrictions.

Reviewers' comments:

Reviewer's Responses to Questions

**Comments to the Author**

1. Is the manuscript technically sound, and do the data support the conclusions?

Reviewer #1: Partly

2. Has the statistical analysis been performed appropriately and rigorously? 

Reviewer #1: Yes

3. Have the authors made all data underlying the findings in their manuscript fully available?

Reviewer #1: Yes

4. Is the manuscript presented in an intelligible fashion and written in standard English?

Reviewer #1: Yes

5. Review Comments to the Author

Reviewer #1: INTRODUCTION

-The author needs to describe the role of Islamic social finance in a broader perspective, not only in Muslim countries, or limited to the COVID-19 pandemic. The illustration can be expanded not only to the problem of the Covid-19 Pandemic (monetary crisis or global economic crisis, global food and financial crisis).

-The author needs to explain what are other supporting factors besides lack of optimal governance, that can be the reasons for the vast gap between the potential of ISF and the reality? Complete with another relevant literature review (LR), such as aspects of Islamic fundraising/finance, management, transparency, mutual trust, etc.

LITERATURE REVIEW

-Literature Review for contemporary Islamic finance needs to be enriched especially from reputable global indexed primary journals, covering Islamic banking, Islamic insurance, etc.

-Provide an explanation of the current state of sustainability of ISF along with references from previous research. If there is no review on this matter, please provide a reference that strengthens your statement that most of the empowerment programs sourced from the ISF do not meet the sustainability aspect.

-The importance of including Literature Review about integration aspects of growth, aspects of justice and aspects of sustainability

METHOD

-The author needs to explain some of the criteria used as input in this research using the two-level ANP method and why the BOCR analysis was selected

-Mention the Literature Review focused on what aspects?

RESULT

-What is the important role in this case? Give a brief explanation

-What institutional aspects are the focus of this article (structure, function/role, coordination system, business partnerships with other parties).

-About integration between Islamic social finance institutions based on capacity development and division of roles, what types of integration are meant (horizontal integration, vertical integration or horizontal coordination, vertical coordination)? Please explain

-Explain what is the difference between short-term and long-term priority solutions shown in table 4 based on the expert agreement?

-It is necessary to conduct a more in-depth analysis and based on this analysis, it is synthesized by the author

DISCUSSION

-Provide adequate arguments, in the financing program covered by sharia financing, success is determined by aspects of distribution, use or utilization (consumptive, productive economy), returns, and governance

-Type of technology, technology level, and what technology transfer and adoption processes will be implemented in managing the instrument of ISF

MANAGERIAL RELEVANCY

-Aspects of human resources that must be improved are technical skills and managerial capabilities. Which of these two aspects is dominant depends on their respective positions.

6. PLOS authors have the option to publish the peer review history of their article (what does this mean?). If published, this will include your full peer review and any attached files.

Reviewer #1: **Yes: **Atika Dyah Perwita

---

## [Author Response · Author response to Decision Letter 0]

13 Feb 2022

We had made the required amendments and changes required by the academic editor and reviewers in the cover letter and in a rebuttal letter labeled 'Response to Reviewers' as attachment. Kindly refer to the attachment files at the main page.

---

## [Decision Letter · Decision Letter 1]

13 May 2022

Integrating Sustainable Islamic Social Finance: Analytical Network Process using Benefit Opportunity Cost Risk (ANP BOCR) Framework: The Case of Indonesia

PONE-D-21-31476R1

Dear Dr. Robani,

We’re pleased to inform you that your manuscript has been judged scientifically suitable for publication and will be formally accepted for publication once it meets all outstanding technical requirements.

Within one week, you’ll receive an e-mail detailing the required amendments. When these have been addressed, you’ll receive a formal acceptance letter and your manuscript will be scheduled for publication. Please note that the reviewer has recommended some language and formatting changes. Please feel free to address these if you wish during this stage.

Kind regards,

Carla Pegoraro

Division Editor

PLOS ONE

Reviewers' comments:

Reviewer's Responses to Questions

**Comments to the Author**

1. If the authors have adequately addressed your comments raised in a previous round of review and you feel that this manuscript is now acceptable for publication, you may indicate that here to bypass the “Comments to the Author” section, enter your conflict of interest statement in the “Confidential to Editor” section, and submit your "Accept" recommendation.

Reviewer #1: All comments have been addressed

2. Is the manuscript technically sound, and do the data support the conclusions?

Reviewer #1: Yes

3. Has the statistical analysis been performed appropriately and rigorously? 

Reviewer #1: Yes

4. Have the authors made all data underlying the findings in their manuscript fully available?

Reviewer #1: Yes

5. Is the manuscript presented in an intelligible fashion and written in standard English?

Reviewer #1: Yes

6. Review Comments to the Author

Reviewer #1: The author has corrected the article properly and completely so that the article is more meaningful to be published in the Plos One journal.

The re-review from the reviewer is only limited to the procedure for writing scientific articles to make it more comfortable for readers to read.

7. PLOS authors have the option to publish the peer review history of their article (what does this mean?). If published, this will include your full peer review and any attached files.

Reviewer #1: **Yes: **Atika Dyah Perwita

---

## [Editor Report · Acceptance letter]

17 May 2022

PONE-D-21-31476R1 

Integrating Sustainable Islamic Social Finance: An Analytical Network Process using the Benefit Opportunity Cost Risk (ANP BOCR) Framework: The Case of Indonesia 

Dear Dr. Robani:

I'm pleased to inform you that your manuscript has been deemed suitable for publication in PLOS ONE. Congratulations! Your manuscript is now with our production department. 

Kind regards, 

on behalf of

Dr Carla Pegoraro 

Staff Editor

PLOS ONE